# The Role of Cytoskeleton Revealed by Quartz Crystal Microbalance and Digital Holographic Microscopy

**DOI:** 10.3390/ijms23084108

**Published:** 2022-04-07

**Authors:** Nicoletta Braidotti, Maria Augusta do R. B. F. Lima, Michele Zanetti, Alessandro Rubert, Catalin Ciubotaru, Marco Lazzarino, Orfeo Sbaizero, Dan Cojoc

**Affiliations:** 1Department of Physics, University of Trieste, Via A. Valerio 2, 34127 Trieste, Italy; nicoletta.braidotti@phd.units.it (N.B.); maria.lima@mr.mpg.de (M.A.d.R.B.F.L.); michele.zanetti@phd.units.it (M.Z.); 2Consiglio Nazionale delle Ricerche (CNR), Istituto Officina dei Materiali (IOM), Area Science Park-Basovizza, Strada Statale 14, Km 163,5, 34149 Trieste, Italy; ciubotaru@iom.cnr.it (C.C.); lazzarino@iom.cnr.it (M.L.); cojoc@iom.cnr.it (D.C.); 3Department of Engineering and Architecture, University of Trieste, Via A. Valerio 6/A, 34127 Trieste, Italy; alessandro.rubert@studenti.units.it

**Keywords:** QCM, DHM, cardiac fibroblasts, cytoskeleton, viscoelasticity, cell rheology

## Abstract

The connection between cytoskeleton alterations and diseases is well known and has stimulated research on cell mechanics, aiming to develop reliable biomarkers. In this study, we present results on rheological, adhesion, and morphological properties of primary rat cardiac fibroblasts, the cytoskeleton of which was altered by treatment with cytochalasin D (Cyt-D) and nocodazole (Noc), respectively. We used two complementary techniques: quartz crystal microbalance (QCM) and digital holographic microscopy (DHM). Qualitative data on cell viscoelasticity and adhesion changes at the cell–substrate near-interface layer were obtained with QCM, while DHM allowed the measurement of morphological changes due to the cytoskeletal alterations. A rapid effect of Cyt-D was observed, leading to a reduction in cell viscosity, loss of adhesion, and cell rounding, often followed by detachment from the surface. Noc treatment, instead, induced slower but continuous variations in the rheological behavior for four hours of treatment. The higher vibrational energy dissipation reflected the cell’s ability to maintain a stable attachment to the substrate, while a cytoskeletal rearrangement occurs. In fact, along with the complete disaggregation of microtubules at prolonged drug exposure, a compensatory effect of actin polymerization emerged, with increased stress fiber formation.

## 1. Introduction

The cytoskeleton is a vital cellular component involved in several processes, and therefore, its dysfunctions can be translated into serious cellular functional impairments [1]. In fact, it was shown that its alterations can be related to diseases, such as some cardiomyopathies [2,3,4]. Among the roles of the cytoskeleton, its contribution to cell rigidity and ability to sense and react to external mechanical stimuli are the most remarkable [2,5]. Moreover, the cytoskeleton is involved in the adhesion process to the external environment through binding proteins, which mediate its connection with the extracellular matrix (ECM) [6]. Adhesion sites are of extreme importance for sensing and signaling the substrate properties changes, both in normal tissue development and pathological conditions. Hence, cell cytoskeletal alterations can be reflected in impaired cell response at the substrate interface [7], with subsequent dysfunctional outcomes. Therefore, this justifies the efforts for a deeper investigation of the cell–substrate interface, both in physiological and altered cell conditions.

Many studies investigated mechanical, viscoelastic, and morphological changes in cells as a result of the effects of some drugs, able to alter cytoskeletal components: microfilaments, intermediate filaments, and microtubules [8,9,10,11,12]. Actin is known to be the main cytoskeletal element responsible for preserving mechanical and morphological properties in healthy cells [5,12,13,14]. It is thus involved in the cytoskeletal reorganization that occurs as a consequence of mechanical signaling or mechanotransduction, which is a process often altered under pathological conditions [2,10]. Microtubules and intermediate filaments seem to have secondary importance but, even if their alteration is not correlated with evident effects at first glance, they are required for the cell mechanical equilibrium [15,16]. For the purpose of cell mechanics investigation, many tools are nowadays employed such as optical tweezers [8], atomic force microscopy [5], and cell stretchers [12]. Among these, there is an emerging interest in the use of quartz crystal microbalance (QCM).

Since QCM allows the detection of minute changes at the near-interface layer with its quartz sensor surface, it has been employed to study cell–substrate adhesion [17,18,19], viscoelastic properties of the extracellular matrix [20], cytoskeletal rearrangements, and cell viscoelastic and morphological changes [21,22,23]. However, the absence of suitable models to predict the behavior of film layers made by cells limits the achievement of quantitative values, allowing only qualitative interpretations of the results. Therefore, the tentative of gaining quantitative mass or viscous information by using QCM is still challenging when working with cells [9,20,22,23]; however, if a complementary quantitative technique is used, the synergic outcomes could be very significant. Among the quantitative tools, digital holographic microscopy (DHM) is a free-label quantitative phase microscopy technique, able to retrieve the height of transparent samples, such as living cells [24,25,26,27,28,29,30]. Thus, quantitative morphological parameters (cell area, volume, and thickness) can be easily obtained, as well as dynamical measurements such as cell membrane fluctuation (*CMF*) [24,27]. 

In this study, QCM was employed for recording changes in frequency (∆f) and dissipation (∆D) values for cells under the action of two cytoskeletal drugs. The near-interface investigation of cellular rheological variations was performed by studying the effect of cytochalasin D (Cyt-D) and nocodazole (Noc), which inhibit actin [14,31] and microtubule polymerization, respectively [32,33]. Afterward, we exploited the advantages of the DHM technique to gain morphological quantitative information on cell area, volume, and thickness.

In this way, the rheological, morphological, and adhesive properties variations due to cytoskeletal alterations were studied by confirming the qualitative interpretations (QCM) with quantitative analysis (DHM).

## 2. Results and Discussion

QCM experiments were performed by using equipment and operation protocols as explained in Section 3. In order to ensure that cell signals (Δf and ΔD) were not hidden by other factors (e.g., viscous properties of the medium and the toxic effect of DMSO on cells), we first performed control experiments (see Section 2.1.3). Subsequently, for confirming the rate of drug action on the cell’s cytoskeleton, we performed actin and tubulin staining. Finally, DHM allowed us to confirm our qualitative deductions by gaining information on quantitative area and volume variations during the treatment.

### 2.1. QCM Results

QCM experiments were performed with different sets of cells under the same seeding and growth conditions. Cell seeding density on the quartz surface and procedures employed are reported in Section 3. After the achievement of steady state, the introduction of the drug solution was carried out and the signals were monitored for four hours. Based on values provided by QCM (Δf and ΔD), the two drugs showed evident differences in rheological response. Control experiments (Section 2.1.3) were performed either with a DMSO-containing medium, in equal volume as that used in the relative drug solution, or separately with the drug solution in the absence of cells. The low content of DMSO on cells did not show any detectable toxic effect, and the only change in liquid properties from the medium to the drug-containing medium was found to be negligible. These lines of evidence guaranteed the possibility of gaining interpretations of the behavior of cells as a direct consequence of the cytoskeletal alterations due to Cyt-D and Noc. 

#### 2.1.1. Frequency Shifts (Δf)

The comparison of frequency shift (Δf) signals obtained for Cyt-D- and Noc-treated cells is reported in Figure 1.

In both curves, we observed initially a fast decrease, due to the activation of the flux for the liquid chamber substitution. After this common feature, the two drug treatments led to independent and different behaviors. Using Cyt-D, the frequency increased and reached the maximum value within a few minutes, comparable with the range of activities on actin [31,34]; then, it was maintained almost constantly in a steady-state-like condition until the end of the experiment. Conversely, using Noc, we observed an initial increase, followed by a gradual decrease for the subsequent two hours. These trends suggest that the rapid loss of cell viscosity due to Cyt-D, observed by other techniques [35,36,37,38], can be also sensed by the quartz sensor. In fact, since the operating condition is outside the limits of Sauerbray’s Equation (2), a positive shift cannot only be assigned to the loss of mass, and Kanazawa’s Equation (3) should also be considered. Moreover, it is known that Cyt-D treatment induces cells to lose the cytoskeleton–integrin–ECM links; thus, some focal adhesions, and consequently the body of the cell retracts, could disappear. This was also observed in human fibrosarcoma [8], NIH3T3, human dermal fibroblasts [21], murine fibroblast-like L929, and the kidney [35,39] cell lines. This suggests that the area in contact with the quartz is highly reduced during the treatment, and even desorption of some cells can occur. 

However, this observation does not agree with the considerations of Tymchenko et al. [21], who claimed that Cyt-D exposure leads to cell body retraction but does not involve changes in the surface area. In order to support our deductions about the decreased area in contact with the quartz surface, we carried out quantitative measurements with DHM (Section 2.3) and observed that this body retraction (cell rounding up) was indeed coupled with a decrease in the projected cell area and an increase in height. Since with our quartz sensor measurement, the penetration depth (δ) of the acoustic wave under water loading was about 180 nm, and assuming that cells have similar properties to those of water, the rounding up action led cells to collocate a major part of their body out of the quartz sensitivity limit. Therefore, the measurement of the cell mass in contact with the quartz decreased and influenced the shift in frequency, as Sauerbray’s equation suggests.

On the other hand, the initial slight increase in Δf observed in Noc-treated cells could be attributed to the tentative restoration of the thermalization value achieved immediately before the solution’s injection. Since the effect of Noc needs more time, compared with that of Cyt-D, for a significant cytoskeletal modification [40,41], the decrease in Δf for the subsequent two hours could reflect the progressive damage of microtubules with increased viscosity [33,42] and subsequent cytoskeletal rearrangement. In fact, meanwhile, the compensatory effect of actin polymerization occurred, emerging from the slight increase in frequency rather than maintaining a firmly steady-state-like condition at prolonged drug exposure. This stimulated effect on actin is well accepted [43,44] and is explained by Noc’s ability to activate the downstream Rho-associated protein kinase (ROCK) pathway, leading to myosin activation and increased actin stress fiber expression, which we were able to confirm qualitatively through immunofluorescence (Section 2.2). 

#### 2.1.2. Dissipation Shifts (ΔD)

Similar to frequency signals, it is possible to observe a clear difference in viscoelastic cells behaviors also in the dissipation profiles (Figure 2).

In Cyt-D, we observed a fast decrease in dissipation, which mirrors the rate of increase in Δf profile. This negative shift can be attributed to the change in cell–substrate adhesion with a transition to a more round-like shape [21,22,34] as a consequence of the Cyt-D effect. In addition, since D corresponds to the ratio between the energy dissipated during the oscillation (G″) and the energy stored (G′) (4), the displacement toward a lower value of D is assumed to be caused by a transition to a more liquid-like cells state [9]. In fact, it was largely demonstrated that, under Cyt-D treatment, both storage and loss moduli decrease [36,45,46]. However, our results show that G″ decreased faster than G′. Since cells are usually characterized by G′ > G″ [36,45,46], at first glance, we found it contradictory. In fact, the increased difference between the two moduli upon treatment suggests a predominant elastic outcoming [47]. Nevertheless, some studies have stated that, since Cyt-D acts on actin filaments, elastic properties of cells, such as Young’s modulus, are reduced upon treatment [36,48,49]. At the same time, it was also shown that the ECM behaves elastically in the absence of significant energy dissipation [20]. Therefore, since the mass in contact with the quartz was reduced upon treatment, it is reasonable to believe that the sensor surface is more sensitive to the larger presence of ECM rather than the presence of cells. In fact, even in the absence of surface precoating, which was not necessary for supporting fibroblast adhesion in our case and was intentionally avoided in order to limit the risk of hiding cell signals, spontaneous adhesion of serum proteins and secretion of extracellular proteins by cells themselves may occur [18].

Conversely, in Noc, we observed a gradual increase in ΔD within the first two hours. This is probably due to the damage of microtubules, resulting in a more viscous behavior, with higher energy dissipation. In fact, during Noc treatment, the tubulin is redistributed through the cytosol [43], as we observed in immunofluorescence experiments (see Section 2.2), and, at this time, cells may be able to oppose more resistance to the quartz oscillation, reflecting an initial increase in the loss modulus (G″). Meanwhile, the actin’s compensatory effect could emerge, becoming preponderant at prolonged drug exposure and resulting in a slight decrease in dissipation during the last two hours, accompanied by an increase in G′ [45].

Unlike the liquid-like behavior and cell detachment observed in Cyt-D-treated cells, under Noc treatment, we did not observe any rounding or loss of adhesion even if a shape reorganization occurred. Despite the Cyt-D effect, is well known that Noc causes a loss in cell polarity [50], accompanied by a redistribution of the surface projected area, which we quantitatively demonstrated with DHM (see Section 2.3). However, the increased loss in viscosity could be attributed to the ability to maintain a firm attachment to the substrate while the reorganization of the cytoskeleton occurs.

#### 2.1.3. Control Experiments

Since the Δf and ΔD shifts caused by changes in viscous properties between medium and drug-containing medium could, in principle, hide cells’ information, control experiments were performed without cells on the quartz, to exclude this hypothesis. Furthermore, even if cells were subjected to a final concentration of DMSO lower than the toxic limit (0.1%), in order to exclude its effect on the behavior of the cells, we performed additional control experiments. For this purpose, as the injecting solution, we used only a DMSO-containing medium by keeping the same volume as that of Cyt-D or Noc used for the drug solution, and then we compared the results. 

As it can be seen from Figure 3 and Figure 4, the frequency and dissipation shifts due to the liquid variation in the absence of cells (no-cell lines) are negligible, compared with those of cells on quartz. The frequency curves resulted in a fast decrease, due to the activation of the flux, which then gradually restored, approaching thermalization values. A similar trend was observed in the dissipation curves. In DMSO lines, we observed a progressive slight increase in frequency signals and a decrease in dissipation. Since post-experimental microscopic observations showed healthy cell morphology, we ascribed the two signals trend to a state of continuous cell spreading. 

In fact, several studies about healthy cells on quartz surfaces show that, under adhesion and subsequent spreading, positive frequency shifts are observed during the reorganization of the cytoskeleton and formation of focal adhesions [9,22,34]. Even if the spreading appears as the opposite of rounding up, we need to highlight again that the QCM’s signals depend on the contributions of different factors, which are not separable. Thus, qualitative interpretations were made, but the weight of a single contribution to the whole formula could not be determined. With this in mind, a positive frequency shift could be attributed to a reorganization of the cytoskeleton and thus an increase in G′ [34,51], while the higher sensed mass could be responsible for the delay resulting in a progressive trend instead of a sudden shift, as the one observed in Cyt-D treatment. On the other hand, the relative negative dissipation shift could also probably be attributed to a rise in the spreading and rigidity, which allow cells to better follow the quartz oscillation. This gradual energy dissipation decrease reached values higher than those observed in Cyt-D treatment and lower than those in Noc. This means that a healthy spread cell is more energy-dissipative than a cell with a disrupted actin network, in which the strong association between cytoskeleton, cellular membrane, and ECM is lost [52,53] but remains less dissipative than a cell with disrupted microtubules, which is characterized by a higher viscous behavior. 

These negligible effects enabled us to provide deductions on QCM signals by considering only changes in cell rheological behavior due to the effect of cytoskeletal drugs. 

### 2.2. Immunofluorescence

Immunofluorescence experiments were performed in order to highlight the rate of drugs action on cytoskeletal elements. From the characteristic times observed in QCM signals and known from data provided in the literature, we decided to monitor the effect of Cyt-D at a shorter exposure time than that of Noc. For this purpose, we treated cells for 10 and 30 min with Cyt-D, and 30 min and 2 h with Noc.

Cyt-D showed the ability to achieve substantial F-actin depolymerization into G-actin monomers in a very short time (10 min), without any additional visible effect at prolonged drug exposure (Figure 5).

This result is also reflected in the QCM signals in which we observed a fast change in frequency and dissipation, followed by a stationary trend.

Conversely, 10 μM of Noc treatment showed enough microtubule depolymerization at 30 min, during which some filaments were still present but completely disappeared at prolonged treatment (2 h). In the meanwhile, the compensatory effect of actin emerged with increased stress fiber formation (Figure 6).

This demonstrates that, despite Cyt-D, the absence of a steady-state condition in QCM profiles during the treatment is related to a continuous cytoskeletal rearrangement.

### 2.3. DHM Results

In order to confirm the deductions achieved from QCM results, DHM was employed by using experimental protocols, as explained in “Materials and Methods”. For the multicell analysis, we performed three independent experiments (m = 3) from which we obtained an overall measurement of more than 30 cells for each measured group within the relative drug treatment. We decided to monitor different intervals of drug exposure by considering the rate of the drug’s action on cytoskeletal biopolymers accessed by QCM and immunofluorescence. Thus, 5 μM Cyt-D treatment exerted a rapid effect on actin with a high contribution within 10 min, while in 10 μM Noc treatment, we observed a more time-dependent and progressive effect on cells. For this purpose, we analyzed treated cells at five different time points, which were 0, 10, 20, 40, and 60 min for Cyt-D and 0, 1, 2, 3, and 4 h for Noc.

From the numerical calculation after hologram reconstruction, we observed a progressive reduction in the projected area, due to Cyt-D, with more than 30% of reduction in the first 10 min and an overall decrease of 65% in 60 min of treatment (Figure 7a). Moreover, we observed an almost constant volume during the entire treatment period (Figure 7b).

Conversely, in Noc treatment, we did not observe any evident morphological variations (Figure 8), even if prolonged drug exposure was monitored.

Noc is known to act on fibroblasts cell shape with the displacement toward a more symmetric cell morphology with less polarity [50]; this is likely accompanied by a redistribution of the surface projected area, which led us to observe constant quantitative morphological trends.

These results confirmed the adhesive properties of the cells during the treatment and our morphological considerations about QCM signals.

In addition, the decreased area along with constant volume observed in Cyt-D treatment led us to consider a possible increase in cell thickness. For this reason, in order to achieve a complete morphological characterization, by exploiting the 3D quantitative information provided by DHM, we decided to delve deeper into the exploration of this treatment. For this purpose, we performed a single-cell analysis, as explained in “Materials and Methods”. Time-lapse experiments (*n* = 5) were able to confirm the results of the multicell analysis with a higher relevance in the real-time monitoring, consistent with the QCM approach.

In Figure 9a, an example of the cells’ hologram reconstruction corresponding to 0 and 15 min of Cyt-D treatment is shown.

For each time lapse, the projected surface area (PSA) and volume (V) were calculated, as explained in “Materials and Methods” for holograms corresponding to 0, 3, 6, 9, 12, and 15 min. Moreover, the mean height (V/PSA) was plotted in order to highlight the increase in thickness.

We were able to appreciate the increase in thickness also from the retrieved 3D images after reconstruction, and for comparing thickness profiles at different time points, we used MATLAB’s Improfile function. We plotted the cell-thickness profiles by choosing a line that crossed the cell in a region where the retraction obviously occurred. In Figure 9b, it is possible to observe cell profiles for t = 0 min and t = 15 min relative to the red line shown in Figure 9a. 

For all time lapses, we observed comparable results as the standard deviations suggest in Figure 10, in which the values for the mean percental variations in the area, volume, and mean height are presented. For plotting the results, the percental variations were calculated from the difference between the final and initial value, normalized to the initial one, and multiplied by 100 as follows:(1)$$\%\;variation=\frac{x_{f}-x_{i}}{x_{i}}*100,$$
where *x* corresponds to PSA, V, or V/PSA (mean height); *i* is the initial value, and *f* is the final value. 

Moreover, we noticed clear comparability in the values extracted from all of the experiments performed in both multicell and single-cell analyses. In summary, we observed a decrease of about 30–40% in the projected area within the first 10 min of treatment and an overall decrease of about 50% in 15 min. This means that, in the last 45 min of treatment, the reduction was only about 25%. Therefore, this proved that the main contribution, due to Cyt-D, was verified within the first 15 min. with a larger effect during the first 10 min. Moreover, by monitoring single cells, we were also able to calculate the cell’s mean height by normalizing its volume upon the projected area, again marking the tendency of the cell thickness to increase over the treatment time.

Despite the morphological investigation, dynamical analysis was unable to provide us with quantitative evaluations. Since cell fluctuations (*CMF*) were in the instrumental noise range, we did not observe any dynamical variations for both treatments.

In conclusion, from quantitative phase information, we confirmed the morphological variation occurring during the Cyt-D treatment, which justifies the reduction in the mass in contact with the quartz surface and related effects on Δf and ΔD. In order to better understand QCM’s frequency shifts, taking advantage of DHM quantitative information, an approximative estimation of the mass variation, occurring under Cyt-D action, was performed. We evaluated the mass variation sensed by quartz, within the penetration depth, due to area variation. For this calculation, values from the multicell analysis were used. The mean area value of untreated cells was considered, as well as that of treated cells at the end of one hour of treatment, which was presumably maintained constantly for the subsequent hours since no relevant changes in Δf and ΔD were observed. We believe that, if Sauerbray’s equation is only considered, such a variation in mass should lead to a higher frequency shift (about 1300 Hz) than that observed; this demonstrates that Kanazawa’s contribution must be involved in the interpretation of our results. Conversely, if Kanazawa’s equation is only considered, an estimation of treated cells’ viscosity could be achieved by considering the viscosity of a healthy cell. For example, by using the apparent viscosity of healthy 3T3 fibroblasts measured with QCM by Wegener et al. [17], we obtained a decrease of about 13% in viscosity after Cyt-D treatment. 

However, during the treatment, we observed a morphological variation in cells in which the rounding up led cell bodies to be mostly outside the penetration depth of the acoustic wave. For this reason, the thus calculated decrease in viscosity is mainly influenced by the presence of medium rather than cells and cannot be interpreted as the real viscosity of cells themselves.

Despite the tentative assumption of simplifications for achieving quantitative information, under our working conditions, the elastic (Sauerbray) and viscous (Kanazawa) contributions cannot be decoupled, as previously stated. Even if QCM’s signals depend on entities of different nature, such as mass, density, viscosity, storage, and loss modulus, there are currently limitations in modeling viscoelastic films composed of live cells, preventing us from the quantitative estimation of these parameters. Thus, unlike quantitative DHM analysis, only qualitative evaluation can be achieved from our QCM’s results.

## 3. Materials and Methods

### 3.1. Cell Culture

Post-natal day 4/5 neonatal Wistar rats were euthanized by decapitation. To isolate cardiac fibroblasts, the hearts were extracted from the abdominal cavity and placed in CBFHH (calcium and bicarbonate-free Hank’s Buffer with HEPES) supplemented with 10 U/mL heparin (H3149, Sigma-Aldrich, St. Louis, MO, USA) and stored on ice. After extraction, fibroblasts were cultured at 37 °C in a 95% H_2_O and 5% CO_2_ atmosphere in Dulbecco’s modified Eagle’s medium (DMEM), high glucose, GlutaMAX^TM^, pyruvate (Thermo Fisher Scientific, Waltham, MA, USA), and supplemented with 10% fetal bovine serum (FBS) (Sigma-Aldrich, St. Louis, MO, USA) and 1% antibiotic–antimycotic (Thermo Fisher Scientific, Waltham, MA, USA). Fibroblasts were used at P0/1 either from fresh cultures, or liquid-nitrogen-stored after previous extractions. 

For QCM measurements, cells at a density of 20,000 cells/cm^2^ were drop-seeded on the surface of the sterilized Au-coated quartz sensor. Afterward, cells were allowed to attach to the incubator for 24 h. 

For DHM experiments, 24 h before measurements cells were seeded into 35 mm Petri dishes at a density of 2600 cells/cm^2^, in order to obtain enough isolated cells. 

### 3.2. Drug Solutions

Cytochalasin D (Cyt-D) (C8273, Sigma-Aldrich, St. Louis, MO, USA) and nocodazole (Noc) (M1404, Sigma-Aldrich, St. Louis, MO, USA) were used to depolymerize actin and microtubules, respectively. Stock solutions were made by dissolving the drugs in DMSO (Sigma-Aldrich, St. Louis, MO, USA) at 4 mg/mL (Cyt-D) and 3.88 mg/mL (Noc). On the day of treatment, the stock solutions were diluted directly inside the medium to a final working concentration of 5 μM (Cyt-D) and 10 μM (Noc), with less than 0.1% of DMSO content. Since the tests were performed in air, 25 mM of HEPES (Thermo Fisher Scientific, Waltham, MA, USA) was used to keep the proper pH balance.

### 3.3. Quartz Crystal Microbalance (QCM)

The QCM’s sensor consists of a thin AT-cut quartz disc on which two gold electrodes are evaporated. An alternating current is applied to the quartz and, due to piezoelectric properties, causes its oscillation, whose frequency is sensible to the amount of the adsorbed mass.

In the case of very thin and completely elastic films, the frequency shift is proportional to the mass in contact with the quartz per unit of electrode’s area. The equation that rules this relationship is known as Sauerbray’s equation [54], which is
(2)$$\Delta f_{1}=-\frac{1}{C}\Delta m,$$
where C is the quartz sensitivity, f is the frequency, and m is the mass per unit of electrode surface area.

However, for such films immersed in liquid, an additional frequency shift is recorded due to liquid properties (density and viscosity). This term is called Kanazawa’s contribute [55], and it correlates viscous properties with frequency shifts as follows:(3)$$\Delta f_{2}=-f_{0}^{\frac{3}{2}}\sqrt{\frac{\rho_{l\;}\eta_{l\;}}{{\pi \mu }_{q}\rho_{q}}},$$
where f_0_ is the unloaded resonance frequency; μ_q_ and ρ_q_ are, respectively, the shear modulus and density of the quartz crystal; ρ_l_ and η_l_ are the density and viscosity of the liquid.

In this way, the recorded total frequency shift consists of the sum of the contributions due to the mass and those due to liquid properties [51].
(4)$$\Delta f=\Delta f_{1}+\Delta f_{2}=-\frac{1}{C}\Delta m-f_{0}^{\frac{3}{2}}\sqrt{\frac{\rho_{l\;}\eta_{l\;}}{{\pi \mu }_{q}\rho_{q}}}.$$

Unfortunately, this equation does not well predict the case of soft films immersed in liquid [51,56]. Moreover, additional complications emerge when the film is not continuous, homogeneous, or even morphologically stable over time, making this technique unable to provide quantitative measurements when working with cells [9,20,22,23]. Thus, we used Equation (4) to achieve qualitative interpretations of our cell layer behavior. Additionally, in the case of liquid loading or viscoelastic films, a reasonable amount of dissipation of the quartz vibrational energy is observed [57,58]. This dissipation (D) is proportional to the energy lost (G″) and stored (G′) during one cycle of oscillation, and it follows the equation
(5)$$\Delta D=\frac{G^{\prime \prime }}{\;G^{\prime }\;2\pi }.$$

Finally, with these two parameters (∆f and ∆D), provided by QCM, it was possible for us to investigate what occurs close to the surface of the quartz at the cell–substrate interface, considering that the penetration depth of the acoustic wave is given by
(6)$$\delta =\sqrt{\frac{\eta }{\pi f\rho }},$$
where η and ρ are, respectively, the viscosity and density of the liquid, and f is the frequency [55,57].

The QCM device used for the experiments in this study was built by Novaetech Srl (openqcm.com), Pompei (NA, Italy). All of the measurements were performed with an AT-cut quartz crystal sensor with a 10 MHz resonant frequency and 11.5 mm diameter gold electrode, placed in the QCM’s chamber of 30 μL in volume. The quartzes used were characterized by a sensitivity (C), shear modulus (μ_q_), and density (ρ_q_) of, respectively, C = 4.42 ng/Hz·cm^2^, μ_q_ = 2.947 × 10^11^ g/cm·s^2^, and ρ_q_ = 2.648 g/cm^3^. The experiments were performed at 37 °C in an oven (M 40–TB, Tecnovetro Srl, Monza, Italy); thus, we could approximate the QCM acoustic shear wave decaying into the liquid to δ = 180 nm, which corresponds to that of a 10 MHz AT-cut quartz crystal under water loading at 37 °C [23]. 

The quartz, with attached cells, was positioned in sterile conditions in the chamber, which was closed and subsequently filled with medium using a tubing system connected to a peristaltic pump (Masterflex C/L^TM^, Cole Parmer, Vernon Hills, IL, USA). Each experiment began with a first calibration phase with medium-only until steady state, when an equilibrium in the temperature profile was observed.

Afterward, the drug-containing medium was fluxed inside the chamber. Once all of the liquid was replaced, the flux was stopped, and the frequency (f) and the dissipation (D) values were continually monitored for the subsequent four hours at intervals of 0.66 s.

### 3.4. Digital Holographic Microscopy (DHM)

The DHM principle, and related codes used for data processing, are based on the optical phase (OP). Quantitative information is retrieved from the OP shift, which is correlated to the optical path difference (OPD) [24] as follows:(7)$$\mathrm{OP}=\frac{2\pi }{\lambda }\mathrm{OPD},\;\mathrm{OPD}=h\;*\;\left(n_{c}-n_{m}\right),$$
where λ is the wavelength of the laser beam, h is the cell thickness, n_c_ is the refractive index of the cell, and n_m_ is that of the medium.

The optical phase shift (OP) between the reference and object beams is contained in the recorded hologram. In contrast to classical holography, digital holography uses a digital camera to record the holograms, which are then numerically reconstructed to calculate the OP shift [25,30]. Knowing the refractive index of the cell (n_c_) and that of the medium (n_m_), the cell height (h) can be easily derived from Equation (7) [24,28,30].

Subsequently, cell projected surface area (PSA), volume (*V*) [24], and cell membrane fluctuation (*CMF*) [27] can be also calculated as follows:(8)$$\mathrm{PSA}=N*p_{a},$$
(9)$$V=p_{a}*\sum_{i=1}^{N}h_{i},$$
(10)$$CMF=\frac{1}{N}\sum_{i=1}^{N}STD_{i},$$
where *N* is the number of pixels within the projected surface area, *p_a_* is the area of a single pixel of the CMOS sensor in the object plane, and *STD_i_* is the phase standard deviation for each pixel. 

In this study, DHM in off-axis configuration, based on a Mach–Zehnder interferometer, was used [24,25,27]. The laser beam (λ = 630 nm, 05-LHP-151, Melles Griot, Bensheim, Germany) was split into object and reference beams and recombined by a cube beam splitter (BS079, Thorlabs Inc., Newton, NJ, USA) to generate the hologram, which was recorded on an sCMOS camera (CS2100M-USB Quantalux, Thorlabs Inc., Newton, NJ, USA). The samples were exposed to a power (P) of the laser set to P ≈ 0.2 mW and the exposure time (t) was 0.5 < t < 1 ms. The magnification of the microscope was set at 33.2X, with a lateral spatial resolution d ≈ 600 nm. An aspheric lens (C230TMD-A, Thorlabs Inc., Newton, NJ, USA) with the numerical aperture NA = 0.55 was used as the objective lens. The size of the sCMOS sensor was 1920 × 1080 pixels (4.8 × 4.8 μm per pixel).

The digital procedure applied to holograms, in order to achieve the reconstruction of the optical phase, was previously reported by our group [24,30]. DHM was employed for monitoring area and volume variation by achieving information from the reconstruction of a single full-frame (1920 × 1080 pixels) image. Cells were segmented, in order to separate them from the background, by manual segmentation of the reconstructed optical phase image, using Image Segmenter, a MATLAB built-in environment. Cell height, area, and volume were calculated, as already explained in [24], considering the refractive index of cell and medium, respectively, as n_c_ = 1.37 and n_m_ = 1.35 [59,60]. Moreover, cell membrane fluctuations (*CMF*) were achieved by using a customized algorithm based on the approach developed by Rappaz et al. [27], which was applied on at least 2 s of a movie recorded with a window of 480 × 290 pixels and a frame rate of 110 frames per second (FPS).

Two types of experiments were performed: multicell and single-cell analyses. In multicell experiments, for each drug treatment, five groups of cells were tested considering the effective range of the drug’s action on cells. For Cyt-D, we analyzed untreated cells and cells after 10, 20, 40, and 60 min of Cyt-D 5 μM treatment. For 10 μM Noc treatment, we considered the following intervals: 0, 1, 2, 3, and 4 h. For each group, we obtained an overall measurement of more than 30 cells for which area, volume, and *CMF* were calculated, as previously explained. Instead, for the second type of experiment, we monitored single cells’ area and volume variations with time-lapse performed at 2 FPS for 15 min, starting immediately after the Cyt-D injection. Before the measurements, each sample was washed with PBS 1X and subjected to medium substitution with a fresh one, for both untreated and time-lapse samples, or to drug solution, for treated cells.

### 3.5. Fluorescence Microscopy

Epifluorescence imaging was performed to observe the cytoskeletal rearrangements. Cells were seeded on round glass coverslips (18 mm diameter) for 24 h before being treated with the desired drug’s solution. For both treatments, one glass was left untreated and used as a negative control. Treated samples were fixed (paraformaldehyde 4%, 20 min) after 10 and 30 min of incubation in Cyt-D or after 30 min and 2 h of incubation in Noc.

After being permeabilized with Triton X-100 0.1% and blocked with bovine serum albumin (BSA) 0.5% for 5 and 30 min, respectively, fixed cells were incubated with the following antibodies or staining solutions: primary antibody to α-tubulin (ab7291, Abcam, Cambridge, England, 1:500), secondary antibody conjugated to Alexa 488 (ab150113, Abcam, Cambridge, England, 1:500), TRITC-conjugated phalloidin (90228, Millipore, Burlington, MA, USA, 1:250) and DAPI (90229, Millipore, Burlington, MA, USA, 1:1000). 

Fluorescent observations were performed using an inverted Axiovert 200M microscope (Carl Zeiss AG, Oberkochen, Germany) upgraded with the appropriate excitation–emission filters for TRITC, FITC, and DAPI spectra, and coupled with a 63X/1.4 Plan Apo oil immersion objective (Carl Zeiss AG, Oberkochen, Germany) as well as an X-cite^®^ 120Q fluorescence illuminator (Excelitas Technologies Corp., Waltham, MA, USA). Images were taken with an XM10 monochrome CCD camera (Olympus Corporation, Tokyo, Japan), connected to a 0.63X adaptor tube, at an exposure of 5 s (FITC and TRITC) or 1 s (DAPI) and by averaging 5 frames for noise reduction. Subsequent image processing was performed using the open source Fiji image processing package (https://imagej.net/software/fiji/, accessed on 1 March 2022).

## 4. Conclusions

We used a new combination of approaches to achieve information on the near interface cellular dynamical variation. The use of two complementary techniques allowed us to overcome the complications related to the use of QCM in the cellular investigation. The lack of models for determining the cell layer behavior on the oscillating quartz allows for only qualitative interpretations. Quantitative DHM was used in this study to confirm the deductions by gaining information on morphological parameters such as area, volume, and cell thickness. We used and compared the effects of two cytoskeletal drugs able to interact with actin and microtubules. QCM was employed for studying the cell–substrate interface changes as a result of the effects of 5 μM Cyt-D and 10 μM Noc treatments for four hours, by measuring and analyzing the Δf and ΔD signals.

In agreement with a typical range of Cyt-D action, confirmed also by immunofluorescence, we observed consistent Δf and ΔD changes within the first 10 min after medium substitution. During the first hour, maximum values were reached, which were maintained almost constantly until the end of the experiment. This suggested the ability of QCM to detect the loss of cell–substrate adhesion and also decreased viscosity. Morphological variations were confirmed by quantitative DHM, which highlighted again the main contribution of Cyt-D within the first 10 min, without significant additional effect at prolonged exposure.

Despite Cyt-D, we did not observe any quantitative morphological variations in Noc treatment. This led us to the conclusion that QCM signals, in this case, can be interpreted as a direct result of cytoskeletal rearrangement rather than cell mass variations. In agreement with immunofluorescence, we observed a time-progressive effect on cells and thus on rheological properties. These results involved a double-step trend with the first redistribution of tubulin through the cytosol, with increased viscous behavior (increased G″), followed by the actin’s compensatory effect and stress fiber formation, which led to a slight decrease in dissipation (increased G′) at prolonged drug treatment.

We should note that there are relatively few reports on the successful application of QCM for live-cell studies. This could be due to the complications correlated to the execution of experiments, as well as the interpretation of their results. To our knowledge, this is the second study in which QCM real-time monitoring is used with primary cultures and the first in which primary cardiac fibroblasts are employed. Moreover, the combination of the two approaches allowed us to compare QCM results to the quantitative DHM technique, achieving synergic and complementary interpretation of cells behavior, with a focus on the near-interface layer.

In conclusion, we studied the rheological, morphological, and adhesive changes in primary cardiac fibroblasts treated with two known cytoskeletal drugs: Cyt-D and Noc. We were able to highlight the pivotal role of actin in maintaining shape integrity, adhesion stability, and mechanical structure. In fact, we observed that in its absence (by using Cyt-D), the cell’s ultimate condition is marked by a loss of adhesion, with the highest possible detachment from the surface and liquid-like behavior. Conversely, cells devoid of microtubules are not involved in a compromised shape or adhesion and, as a result of a counterbalance play, their cytoskeleton emerged with increased actin stress fibers. This last line of evidence, reported also in other studies, could be justified by considering the Tensegrity model’s principles for which the cell’ mechanical stability is guaranteed by a tensile prestress [15,16].

The viscoelastic and morphological changes studied again reveal the importance of the right understanding of cellular reorganization that occurs when cells are subjected to defects. 

Cytoskeletal alteration is known to be the cause of many diseases, such as cardiomyopathies. Even if our approach is relatively far from direct use in disease diagnosis, the lack of comprehensive knowledge regarding biophysical cell behavior needs to be filled, which is planned for future research. Therefore, a deep understanding of the interconnection between alterations in nano-/microstructures and macroscopic behavior plays a central role in the development of future powerful biomarkers of mechanical nature for diagnostic purposes. 

## Figures and Tables

**Figure 1 ijms-23-04108-f001:**
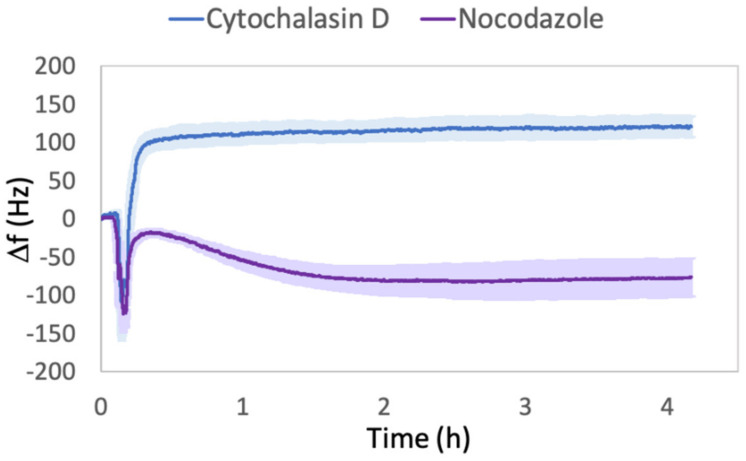
Comparison between frequency shifts (Δf) of Cyt-D- and Noc-treated cells. The mean values are relative to three independent replicas performed with different sets of cells. The reported Δf values are relative to f signals recorded from 10 min before the drug solution injection (Cyt-D 5 μM or Noc 10 μM) until the end of four hours of treatment.

**Figure 2 ijms-23-04108-f002:**
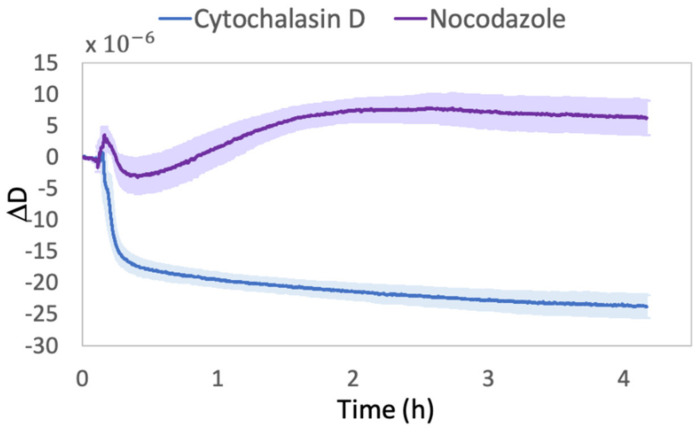
Comparison between dissipation shifts (ΔD) of Cyt-D- and Noc-treated cells. The mean values are relative to three independent replicas performed with different sets of cells. The reported ΔD values are relative to D signals recorded from 10 min before the drug solution injection (5 μM Cyt-D or 10 μM Noc) until the end of four hours of treatment.

**Figure 3 ijms-23-04108-f003:**
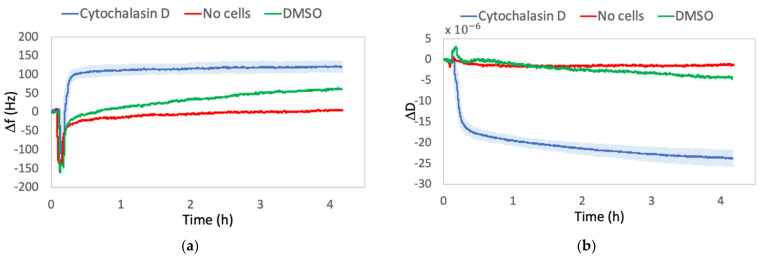
Comparison between Δf (**a**) and ΔD (**b**) shifts in the experimental condition (Cytochalasin D) and in control condition: only drug solution in the absence of cells (No cells) or DMSO-containing medium in equal quantity as that one used for the drug solution (DMSO). The reported Δf and ΔD values are relative to f and D signals, recorded from 10 min before the solution injection until the end of four hours of treatment.

**Figure 4 ijms-23-04108-f004:**
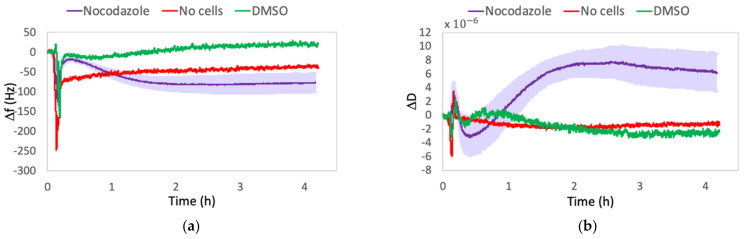
Comparison between Δf (**a**) and ΔD (**b**) shifts in the experimental condition (Nocodazole) and in control condition: only drug solution in the absence of cells (No cells) or DMSO-containing medium in equal quantity as that one used for the drug solution (DMSO). The reported Δf and ΔD values are relative to f and D signals, recorded from 10 min before the solution injection until the end of four hours of treatment.

**Figure 5 ijms-23-04108-f005:**
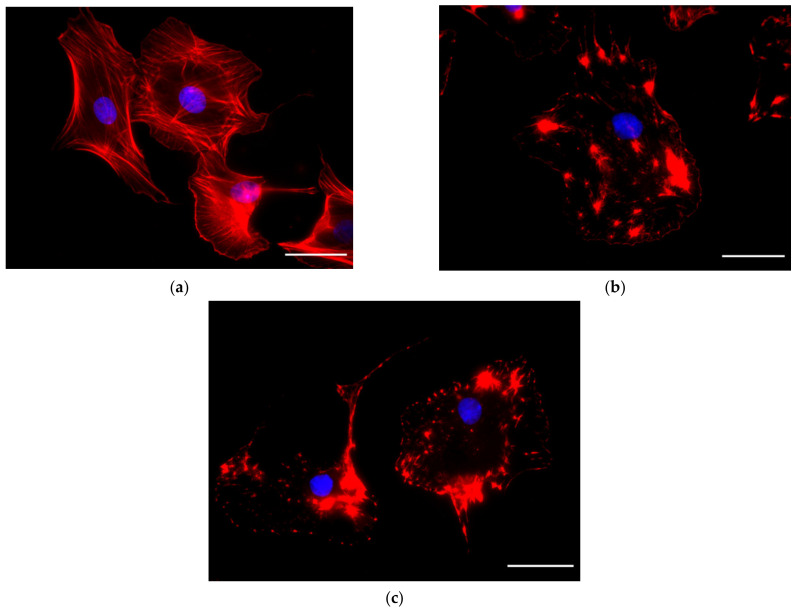
Effect of Cyt-D 5 μM on cytoskeletal actin. Fluorescent images for control (**a**) and cells after 10 min (**b**) and 30 min (**c**) of treatment. Scale bar: 40 μm.

**Figure 6 ijms-23-04108-f006:**
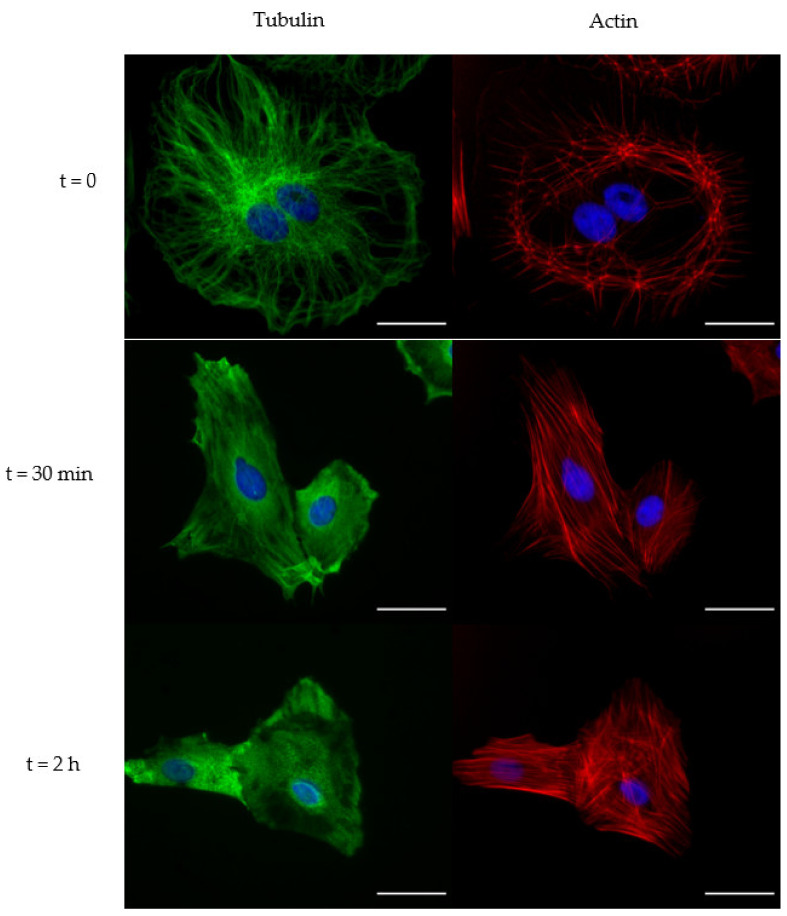
Effect of 10 μM Noc treatment on cytoskeletal tubulin and actin. Fluorescent images for control (t = 0) and cells after 30 min (t = 30 min) and 2 h (t = 2 h) of treatment. Scale bar: 40 μm.

**Figure 7 ijms-23-04108-f007:**
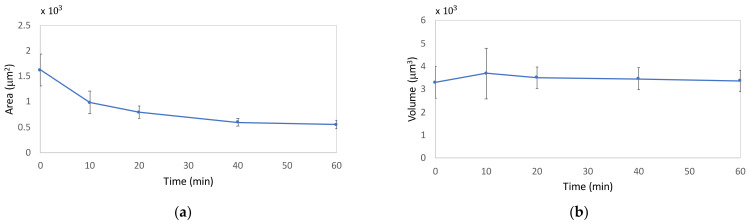
Morphological parameters variations due to Cyt-D at 0, 10, 20, 40, and 60 min. Reported mean values and standard deviations for area (**a**) and volume (**b**) are relative to three independent replicas (m = 3).

**Figure 8 ijms-23-04108-f008:**
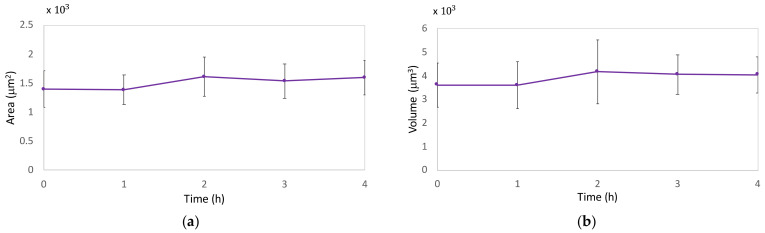
Morphological parameters variations due to Noc at 0, 1, 2, 3, and 4 h. Reported mean values and standard deviations for area (**a**) and volume (**b**) are relative to three independent replicas (m = 3).

**Figure 9 ijms-23-04108-f009:**
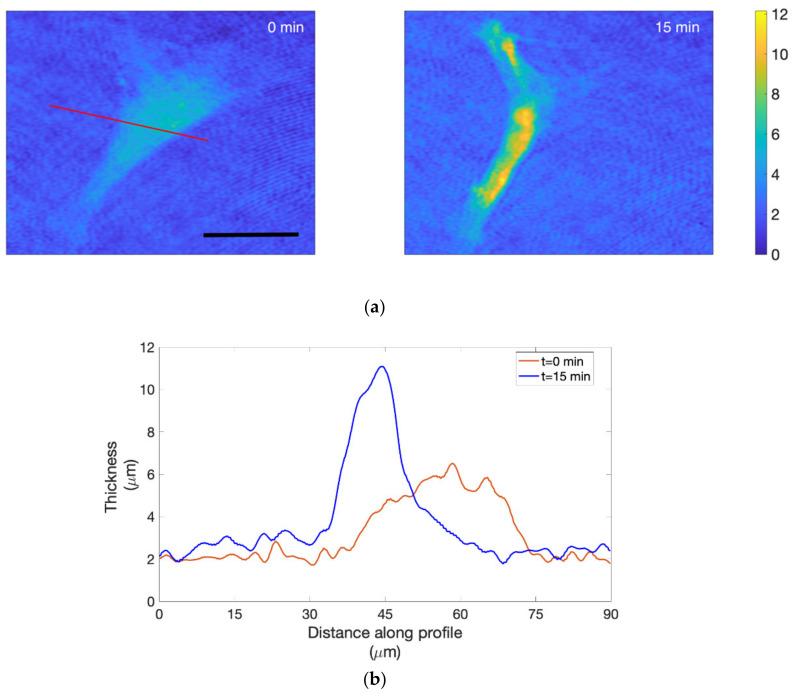
Thickness map (μm) and cell-thickness profiles: (**a**) thickness map (μm) of reconstructed holograms extracted from the time lapses at 0 and 15 min. Scale bar: 50 μm; (**b**) cell-thickness profiles relative to t = 0 min (orange line) and t = 15 min (blue line) were plotted along the red line across the cell.

**Figure 10 ijms-23-04108-f010:**
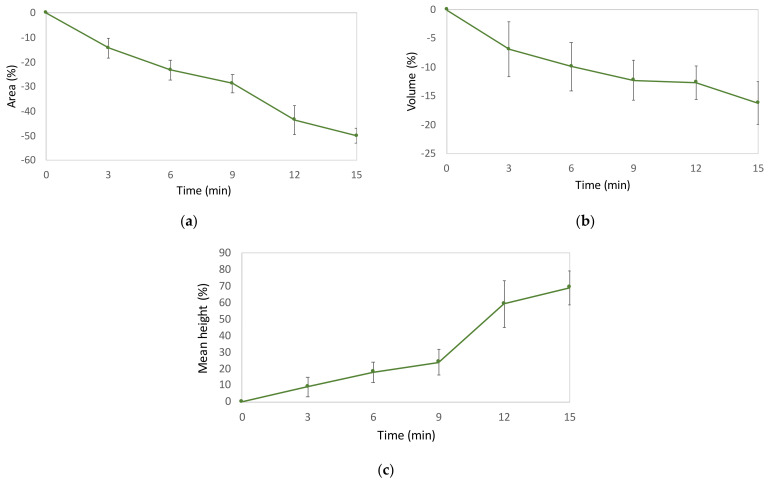
Percental variations in area (**a**), volume (**b**), and mean height (**c**) obtained in single-cell experiments (*n* = 5) relative to 0, 3, 6, 9, 12, and 15 min of Cyt-D treatment.

## Data Availability

All authors confirm that all related data supporting the findings of this study are given in the article.

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
