# Peer review of "The Role of Cytoskeleton Revealed by Quartz Crystal Microbalance and Digital Holographic Microscopy"

_ijms, 2022, doi:10.3390/ijms23084108_

Round 1
Reviewer 1 Report
The work described here has the novelty of applying Quartz Crystal Microbalance (QCM) and Digital Holographic Microscopy (DHM) to gather information on cell viscoelasticity and adhesion changes at the cell-substrate near-interface and morphological changes due to the cytoskeletal alterations, respectively, upon treatment with cytoskeleton-modifying drugs Ctytochalasin D and Noconazole.
While the authors do provide the appropriate DMSO-treated control in the supplementary data, the same crucial control is lacking in all the main data of the manuscript. Since the drugs used here used DMSO as a vehicle it is impossible to assert that any of the effects observed were not influenced by the vehicle itself. Therefore, all the results shall be revised to include this critical control.
Author Response
We carefully considered the referees’ comments and adjusted the paper accordingly.
The main change consisted in the insertion of the control experiments into the main text in order to clarify the aspects highlighted by the first reviewer. In the first submission, control experiments were placed in Supplementary Materials with the intention to avoid disruption of the main text flow. In order to better demonstrate that they are, indeed, the controls of the experiments performed and reported in the main text, we decided to place them inside the “Results and Discussion” section (3.1.3. Control experiments).
Reviewer 2 Report
This manuscript describes the effect of two drugs known to disrupt the actin and tubulin cytoskeleton on the physical properties of cells. The paper is very well written, and the experimental details are clear. A sophisticated biophysical approach has been taken and the importance of the paper is certainly not the biological results obtained as all findings reported here have been known for decades, rather it is the techniques used that are the interest here. This being the case I think that the title should reflect the technology and approach used rather than focussing on these ancient findings? Something like “The role of the cytoskeleton revealed by Quartz Crystal Microbalance and Digital Holographic Microscopy.”
Does the volume of cells actually decrease after Cyt D treatment as suggested by figure 8? Or (more likely) does the ability of the technique to measure the intricate shape that rounded cells adopt (huge numbers of fine filopods and other cell processes) fail to measure cell volume accurately?
Small specific points
Line 209 Perhaps 18mm diameter would be clearer than 18mm ø?
Figure 5 and 6. The legend should indicate if the error bars are SD or SE.
The concluding paragraph starting on line 555 is not at all convincing. I don’t think there is any real prospect of using these techniques to diagnose cardiomyopathies!
Author Response
According to your comments, we agree to the proposed title: “The role of the cytoskeleton revealed by Quartz Crystal Microbalance and Digital Holographic Microscopy” which fits better the article intentions. Moreover, we took care of the minor suggestions and we promptly provided clarifications or corrections inside the main text.
About the volume values of Figure 8 (which in the updated manuscript corresponds to Figure 10) we feel the necessity to clarify the reviewer’s doubt.
The mentioned figure is relative to single-cell time lapse where we observed a relatively small change in volume if it is compared with the surface project area variations. We observed just a small percental decrease in volume with a final value of 16 ± 3.75 % of reduction at 15 minutes of treatment. Since the reduction in area was higher (50 ± 3.14 %) we were able to highlight the increased cell thickness by normalizing the volume upon the surface area at each treatment time points. We consider the observed changes in volume values to not being an artefact; in fact, changes due to cell reaction to drugs can occur not only in terms of surface area variation. In the meantime, by performing multi-cell analysis, it emerges that this relative volume variation is not a predominant outcoming among the cells, as it happens, instead, for the area reduction. In conclusion we can consider that the cell volume is just lightly affected by the treatment (and better appreciable by monitoring single-cell with time lapse), while the main influence is on the surface projected area and cell thickness.
We thank again for the consideration and suggestions and, hoping to have satisfied your requests, we look forward to your feedback.
Round 2
Reviewer 1 Report
The supplementary figures included now in the main text allow for better understanding. However in figs 4 and 5 different colors should be used to better distinguish between drug, no cells and DMSO as the figures are still non-intuitive and hard to read.
Figures 1, 2, 5, 6 and 7 are all lacking a DMSO control. Untreated cells are not an appropriate control for these experiments as both compounds used to characterize the changes in cell parameters evaluated here use DMSO as vehicle. It can not be ruled out that DMSO alone does not cause some of the changes claimed to be resulted from drug treatment.
Author Response
We thank again the Reviewer for the attention conferred to our submitted work.
In the updated version of the article, we have modified the figures by changing the colors for a better reading as the Reviewer suggested.
We carefully considered once again the referee’ comments but we feel the need to clarify some aspects related to control experiments.
Figure 1 and Figure 2 show curves obtained with QCM using Cytochalasin D (CytD) and Nocodazole (Noc).
Figure 3 and Figure 4 represent the comparation between curves relative to experimental conditions (the same curves of Figure 1 and Figure 2 are reported) and control conditions (the first one performed in the absence of cells and the second one in the presence of DMSO only). We observed negligible effects on QCM signals, which means that viscous properties variations between medium to drug-containing medium do not influence our measurements. Moreover, also DMSO alone was unable to give rise to significant shifts in frequency and dissipation, which means that it cannot be considered responsible of any possible artefact in our data interpretations.
Figure 5 and Figure 6 are relative to immunofluorescence experiments. With these experiments we would like to establish the effective rate of drug action on cytoskeletal rearrangement. Control experiment in these contexts is usually the negative sample, which is the one not treated with the drug solution but subjected just to a medium change with fresh one, even if these drugs are prepared by dissolving powder in DMSO or other solvents [D. Borin et al., Micron, 102, 2017, 88-96].
Figure 7 is relative to morphological parameters obtained with DHM. In this case, DMSO control is not reported because it was previously demonstrated in the text that the low content used here is unable to cause adverse effects on cells. In addition, we would like to highlight that Noc is used in higher concentrations than CytD, introducing a larger volume of DMSO (although the content remains below the toxic limit) into the sample. Nevertheless, we did not observe any morphological variation due to Noc, in agreement with literature. At the same time, this morphological unchangeable behavior, along with the negligible effect proved with QCM, allows us to consider the DMSO irrelevant for the health of our cells at the doses used.
Furthermore, we are showing below several cells before and after the addition of DMSO at the highest concentration used in our experiments (pictures obtained with atomic force microscopy). Looking at the panel from left to right we have in the first panel the appearance of the culture before adding the DMSO e than two subsequent scans describing the application of DMSO (20 and 40 minutes post-application). It can be seen that the cellular morphology does not change.
We also ran some toxicity tests, here are the results. We evaluated the presence of ATP which indicates the vitality/cellular metabolism. All the numbers are absorbance and/or absorbance ratios:
With DMSO Mean
1 hour 647488 639392 624680 648972 640133
3 hours 883076 890852 912764 916904 900899
6 hours 608872 603692 705572 701332 654867
No DMSO Mean
1 hour 622208 687284 662428 670564 660621
3 hours 886468 940392 932732 895792 913846
6 hours 633020 658652 646276 594660 633152
ASD you can see there are no macroscopic differences
With the above discussions, we hope to have provided you with complete and reasonable explanations. We thank you again for the consideration and suggestions and we look forward to your feedback.
Prof. Orfeo Sbaizero (University of Trieste, Department of Engineering and Architecture, Trieste, Italy) and Dr. Dan Cojoc (Consiglio Nazionale delle Ricerche (CNR), Istituto Officina dei Materiali (IOM), Area Science Park-Basovizza, Trieste, Italy)
Round 3
Reviewer 1 Report
Thank you for the revisions and for the detailed rationale behind the concepts proposed here for how the data is presented.
The additional discussion and contextualization of the data are also appreciated and may help define the contributions of this work to the field.